# Murlentamab, a Low Fucosylated Anti-Müllerian Hormone Type II Receptor (AMHRII) Antibody, Exhibits Anti-Tumor Activity through Tumor-Associated Macrophage Reprogrammation and T Cell Activation

**DOI:** 10.3390/cancers13081845

**Published:** 2021-04-13

**Authors:** Mélissa Prat, Marie Salon, Thibault Allain, Olivier Dubreuil, Grégory Noël, Laurence Preisser, Bérangère Jean, Lydie Cassard, Fanny Lemée, Isabelle Tabah-Fish, Bernard Pipy, Pascale Jeannin, Jean-François Prost, Jean-Marc Barret, Agnès Coste

**Affiliations:** 1UMR 152 Pharma Dev, Université de Toulouse, IRD, UPS, 31062 Toulouse, France; melissa.prat@live.fr (M.P.); marie.salon@wanadoo.fr (M.S.); alaeddine_mohamad@live.fr (T.A.); bernard.pipy@inserm.fr (B.P.); 2RESTORE Research Center, Université de Toulouse, INSERM, CNRS, EFS, UPS, 31100 Toulouse, France; 3GamaMabs Pharma, 31106 Toulouse, France; odubreuil@gamamabs.fr (O.D.); Berengere.jean@pierre-fabre.com (B.J.); fanny.lemee@evotec.com (F.L.); jfprost@gamamabs.fr (J.-F.P.); jmbarret@gamamabs.fr (J.-M.B.); 4Institut Jules Bordet, Université Libre de Bruxelles, 1000 Brussels, Belgium; gregory.noel@bordet.be; 5Univ Angers, Université de Nantes, CHU Angers, Inserm, CRCINA, SFR ICAT, 49000 Angers, France; laurence.preisser@univ-angers.fr (L.P.); pascale.jeannin@univ-angers.fr (P.J.); 6Laboratory of Immunomonitoring in Oncology, Gustave Roussy, 94905 Villejuif, France; lydie.cassard@gustaveroussy.fr

**Keywords:** murlentamab, glyco-engineered antibody, tumor-associated macrophages, ovarian cancer, adaptive immunity, immunotherapy

## Abstract

**Simple Summary:**

AMHRII, the anti-Müllerian hormone receptor, is selectively expressed in normal sexual organs in healthy adults but is also re-expressed in ovarian, colorectal and lung cancers. In this context, we developed murlentamab, a humanized glyco-engineered anti-AMHRII monoclonal antibody, currently in clinical trial. Preliminary data suggest that murlentamab anti-tumor activity involves immune response activation. Thus, in vitro experiments were performed to precisely characterize the murlentamab effect on the human immune system. We show that murlentamab treatment is associated with evidences of innate and adaptive immune cell activation in cancer patient samples. Moreover, we demonstrate that the murlentamab opsonization of AMHRII-expressing ovarian tumor cells promotes a polarization switch of both naïve and tumor-associated macrophages towards an anti-tumor M1-like phenotype. Our work also supports that, through macrophage reeducation, murlentamab activates an anti-tumor adaptive immune response. Finally, the combination of murlentamab with pembrolizumab confirmed novel clinical perspectives of murlentamab association with checkpoint inhibitors and other immuno-modulators.

**Abstract:**

AMHRII, the anti-Müllerian hormone receptor, is selectively expressed in normal sexual organs but is also re-expressed in gynecologic cancers. Hence, we developed murlentamab, a humanized glyco-engineered anti-AMHRII monoclonal antibody currently in clinical trial. Low-fucosylated antibodies are known to increase the antibody-dependent cell-mediated cytotoxicity (ADCC) and antibody-dependent cellular phagocytosis (ADCP) potency of effector cells, but some preliminary results suggest a more global murlentamab-dependent activation of the immune system. In this context, we demonstrate here that the murlentamab opsonization of AMHRII-expressing ovarian tumor cells, in the presence of unstimulated- or tumor-associated macrophage (TAM)-like macrophages, significantly promotes macrophage-mediated ADCC and shifts the whole microenvironment towards a pro-inflammatory and anti-tumoral status, thus triggering anti-tumor activity. We also report that murlentamab orients both unstimulated- and TAM-like macrophages to an M1-like phenotype characterized by a strong expression of co-stimulation markers, pro-inflammatory cytokines and chemokines, favoring T cell recruitment and activation. Moreover, we show that murlentamab treatment shifts CD4^+^ Th1/Th2 balance towards a Th1 response and activates CD8^+^ T cells. Altogether, these results suggest that murlentamab, through naïve macrophage orientation and TAM reprogrammation, stimulates the anti-tumor adaptive immune response. Those mechanisms might contribute to the sustained clinical benefit observed in advanced cancer patients treated with murlentamab. Finally, the enhanced murlentamab activity in combination with pembrolizumab opens new therapeutic perspectives.

## 1. Introduction

The Anti-Müllerian hormone type II receptor (AMHRII), also known as MIS type II receptor (MISRII or MISIIR), is a member of the transforming growth factor beta (TGF-β) receptor superfamily [1,2]. AMHRII plays a major role in male fetus sexual differentiation by inducing, as a consequence of AMH stimulation, the regression of Müllerian ducts, precursors of female reproductive organs (uterus, fallopian tubes and upper vagina) [3]. In adults, AMHRII seems to have a restricted expression profile, being mainly expressed in granulosa cells in women from birth to menopause, acting in follicular growth modulation [4] and expressed in Sertoli and Leydig cells of males with involvement in androgen biosynthesis regulation [5]. Several studies have confirmed the expression of AMHRII in gynecological cancer tissues [6,7,8,9,10] and, more recently, AMHRII was found expressed by certain non-gynecological cancer such as non-small cell lung cancer and colorectal cancer [11,12]. This set of consistent evidence led to the development of murlentamab, also named GM102 or 3C23K, a humanized glyco-engineered anti-AMHRII monoclonal antibody. Following an extensive pharmacological profiling as well as toxicological studies in cynomolgus monkeys [12,13,14], a phase I study of murlentamab in gynecological cancers (www.clinicaltrial.gov/NCT02978755, accessed on: 20 June 2016) and a phase IIa in colorectal cancer (www.clinicaltrial.gov/NCT03799731, accessed on: 11 July 2018) are currently ongoing.

Effector functions mediated by the Fc part of immunoglobulins have been reported to be strongly related to their *N*-linked oligosaccharide structures [15]. Particularly core fucosylated oligosaccharides showed weaker binding to the FcγRIIIa receptor (CD16) expressed on effector cells and resulted in a decreased lytic potential [16,17]. This important therapeutic notion has led to the development of Fc glyco-engineered antibodies, especially low-fucosylated antibodies, for increasing CD16 affinity, for improving antibody-dependent cell-mediated cytotoxicity (ADCC) and antibody-dependent cellular phagocytosis (ADCP) and for compensating the inhibitory effect of the high amount of multi-specific immunoglobulins G (IgGs) from serum of patients [18]. Based on this technology, the anti-CD20 antibody obinutuzumab was the first glyco-engineered antibody approved for chronic lymphocytic leukemia and follicular lymphoma treatment [19,20]. Regarding solid tumors, imgatuzumab, one of the most advanced glyco-engineered antibody, showed promising results in epidermal growth factor receptor (EGFR)-positive solid tumors [21], in head and neck squamous cell carcinoma and in colorectal cancer showing complete and partial response in Phase I and II studies [22,23], especially with tumors highly infiltrated by immune cells at baseline. Interestingly, clinical responses were associated with a robust decrease in peripheral Natural Killer (NK) counts attributed to the migration of NK within the tumor [21,22,23]. This decrease is the main modulation of the immune system reported with glyco-engineered antibodies in clinic up to now.

Although NK cells are the main effector type in both physiological and therapeutic settings, other myeloid types such as dendritic cells (DC), neutrophils and monocytes/macrophages are also involved in ADCC [24]. Because of their noticeable plasticity which allows them to perform several functions within the tumor microenvironment, macrophages represent promising therapeutic targets for cancer treatment. Indeed, accumulating preclinical and clinical observations demonstrated that modulating macrophage polarization in the tumor microenvironment may represent an additional approach for cancer treatment, either alone or in combination with immune checkpoint therapies [25,26]. Macrophages tailor their polarization states and responses depending on stimulatory signals received from the tumor stroma [27,28,29,30,31]. Although it is now clearly recognized that they can adopt a wide spectrum of polarization states [32,33,34], macrophages are generally defined as two extremes: classically activated M1 and alternatively activated M2 cells [35]. M1 macrophages are characterized by the expression of Toll-like receptors (TLRs) and opsonic receptors (e.g., TLR2/4, CD16, CD32, CD64), the production of pro-inflammatory mediators and exhibit a strong anti-tumor activity. Conversely, M2 macrophages express an abundant level of non-opsonic receptors (e.g., CD163, CD36, CD206), produce anti-inflammatory cytokines and contribute in many ways to tumor progression [36]. Higher tumor infiltration of these M2 macrophages, also called tumor-associated macrophages (TAMs), is associated with more aggressive tumor characteristics. Although the precise origin of TAMs is still under debate, circulating monocytes represents well-described TAM precursors, revealing monocyte subsets as critical in the acquisition of pro- or anti-tumor functions of TAMs [37]. Recently, Prat and colleagues showed in patients with ovarian cancer that murlentamab treatment is associated with changes in the proportions of blood monocyte subsets [38]. Furthermore, Bougherara et al. assessed the effect of murlentamab on immune cells and suggested that TAMs could be involved in the anti-tumor potential of this antibody in ovarian carcinoma [14]. In this context, understanding the fine-tuned mechanisms by which murlentamab exerts its anti-tumor activity and investigating its impact on macrophage polarization might provide more arguments in favor of its use in cancer therapy.

In this study we show that murlentamab treatment is associated with several evidences of innate and adaptive immune cell activation in cancer patient samples. Moreover, through an in vitro co-culture system, we demonstrate here that the murlentamab opsonization of AMHRII-expressing ovarian tumor cell lines promotes a shift in polarization of both naïve (M0) and TAM-like macrophages towards an anti-tumor M1-like phenotype. Our work also supports that, through macrophage orientation, murlentamab activates an anti-tumor adaptive immune response. Finally, both in vitro and in vivo studies with murlentamab in combination with pembrolizumab confirmed novel clinical perspectives of murlentamab association with checkpoint inhibitors and other immuno-modulators.

## 2. Results

### 2.1. Murlentamab Treatment of Colorectal and Ovarian Cancer Patients Is Associated with the Activation of the Innate and Adaptive Immune Response

Phase I (C101) then phase IIa (C201) studies were performed with murlentamab to determine recommended doses and treatment responses, characterized by overall survival (OS) and progression-free survival (PFS) (Appendix A). Moreover, these studies allowed to assess pharmacodynamic (PD) markers in patient blood and biopsies that contribute to mechanistically specify the mode of action of murlentamab in human. Thus, patient samples from these two studies were collected, processed and analyzed to determine their cytokinic blood profile by ELISA and the activation of immune cells in both blood and the tumor microenvironment by flow cytometry and immunohistochemistry at baseline and at different time points following murlentamab infusion.

All blood samples of colorectal cancer patients from the Belgian hospital (*n* = 20) were analyzed before (baseline) and during cycle 1 (at D15) and cycle 2 (at D43) of murlentamab treatment. Murlentamab infusion induced both an increase in the proportion of blood monocytes positives for the CD69, an early activation marker, and a decrease in CD69^+^ activated-regulatory T cells (Tregs), whilst no significant variation was noticed regarding CD69^+^ CD8^+^ T cells at these times of blood sampling (Figure 1A). Blood samples from ovarian cancer patients were not centralized but tested in each clinical site, using site-specific markers. Therefore, analysis of blood samples from patients treated at Gustave Roussy (Paris, France) (*n* = 4) also showed signs of immune activation characterized by an increased proportion of CD8^+^ T cells expressing the inducible co-stimulatory molecule (ICOS) in all four patients tested during murlentamab treatment (Figure 1B). In addition, treatment of colorectal cancer patients by murlentamab was associated with an increase in CXCL9 and CXCL10 blood levels, two interferon-inducible chemokines described as critical immune modulators and survival predictors in colorectal cancer [39], in 13 patients out of 16 and 9 patients out of 15, respectively (Figure 1C).

We also evaluated immune markers at the tumor site when patients were volunteers for biopsies before treatment (baseline) and in cycle 2 (at D22). Analysis of eight paired biopsies from C201 showed that murlentamab treatment was associated with a decrease in CD14^+^ cells positive for the CD163 receptor (Figure 1D), a marker generally used to identify TAMs in malignant diseases [40]. Due to the size of biopsies, fewer paired biopsies from C201 could be analyzed with other markers and thus an increase in ICOS^+^ cells was noticed in four out of five patients tested (Figure 1D). In terms of effector cells, four paired biopsies from C201 (Figure 1E) and the two paired biopsies from C101 (Figure 1F) showed that murlentamab treatment induced an increase in the proportion or number of both CD16^+^ cells (in the two patients tested from the C101 study and four out of five patients from C201) and CD16^+^ cells expressing Granzyme B (in the two patients from the C101 study and in three out of four patients from C201), suggesting a murlentamab-induced activation of macrophages and/or Natural Killer (NK) cells into the tumor microenvironment. Moreover, in colorectal cancer patients’ tumor biopsies (C201), murlentamab treatment was associated with increased proportions of CD8^+^ and ICOS co-stimulatory molecule-positive cells in three out of four biopsies tested (Figure 1E).

Altogether, these observations from clinical studies of colorectal and ovarian cancer patients strongly suggest that murlentamab treatment induces a local and systemic activation of both innate and adaptive immune cells.

### 2.2. Murlentamab Enhances Naïve Macrophage and Tumor-Associated Macrophage Intrinsic Anti-Tumor Activity

In order to further characterize the precise mechanisms by which murlentamab could impact immune cell activation and given the pivotal role of macrophages within the tumor microenvironment, we first assessed whether murlentamab could promote and/or restore the in vitro anti-tumor activity of naïve macrophages and TAMs.

For this purpose, SKOV3 human ovarian tumor cells modified to strongly express AMHRII (SKOV3-R2^+^) were opsonized with murlentamab, 3C23K-FcKO (a control antibody mutated in the constant part) or 3C23K-CHO (the 3C23K normal fucose form). These opsonized tumor cells were then co-cultured with naïve unstimulated human monocyte-derived macrophages (MDMs) (M0) or M-CSF/IL-10-treated human MDMs used to mimic TAMs found in the tumor microenvironment [41,42] (Appendix A).

As illustrated in Figure 2A, irrespective of whether the co-culture was done with unstimulated (M0) or TAM-like MDMs, the number of SKOV3-R2^+^ tumor cells opsonized with the 3C23K-FcKO control increased over time. Interestingly, while the opsonization of SKOV3-R2^+^ tumor cells with the 3C23K-CHO did not modify the kinetic of tumor cell number, we observed a significant reduced number of murlentamab-opsonized SKOV3-R2^+^ at D4 in the presence of M0 MDMs and at both D4 and D5 in the presence of TAM-like MDMs with respect to 3C23K-FcKO (Figure 2A). These results suggest that the murlentamab opsonization of ovarian tumor cells promotes and/or restores the direct anti-tumor activity of naïve macrophages and TAMs.

ADCC is a well-described function of macrophages and NK cells that has been attributed to the anti-tumor activity of several antibodies used for cancer immunotherapy [24]. In this context, we explored whether the murlentamab opsonization of ovarian tumor cells could trigger human MDM-dependent ADCC. Interestingly, we showed that opsonization of SKOV3-R2^+^ tumor cells with murlentamab strongly triggered ADCC mediated by both unstimulated (M0) and TAM-like macrophages, whereas 3C23K-CHO was only able to slightly induce ADCC mediated by TAM-like macrophages (Figure 2B). These results demonstrate that the opsonization of tumor cells by murlentamab significantly promote and/or restore the ADCC anti-tumor mechanism in both naïve macrophages and TAMs.

### 2.3. Murlentamab Orients Naïve Macrophages and Reprograms Tumor-Associated Macrophages towards an M1-Like Profile

Following the previous results regarding clinical studies and considering the study of Bougherara et al. implying an activation of T cells through MDMs in the murlentamab-induced killing of tumor cells [14], we further evaluated in vitro whether the murlentamab opsonization of human ovarian tumor cells could impact macrophage phenotype.

After 3 days of co-culture with opsonized SKOV3-R2^+^ tumor cells, the proportion of MDMs expressing M1 or M2 markers was evaluated by flow cytometry (see Table 1 for antibodies used) and their ability to produce pro- and anti-inflammatory mediators was assessed by AlphaLisa immunoassays (Appendix A).

Regarding Fc-gamma receptor M1 polarization markers, almost all unstimulated (M0) and TAM-like MDMs expressed the CD32 receptor whatever the culture conditions (Figure 3A). Hence, the murlentamab treatment did not significantly impact this high proportion of CD32^+^ M0 and TAM-like MDMs (Figure 3A). However, SKOV3-R2^+^ opsonization with 3C23K-CHO and murlentamab increased the proportion of TAMs expressing the CD64 receptor as compared with the opsonization with the 3C23K-FcKO (Figure 3A). Moreover, the co-culture with SKOV3-R2^+^ cells opsonized with 3C23K-CHO and murlentamab also resulted in a strong induction of the proportion of cells expressing the CD80 co-stimulation molecules and the TLR2 receptor in both M0 and TAMs (Figure 3A). Mirroring this increase in cells expressing M1 markers, the co-culture with SKOV3-R2^+^ cells opsonized with 3C23K-CHO and murlentamab decreased the proportion of M0 MDMs expressing CD163, CD36 and CD206, three M2 markers involved in TAM tumor promoting and immunosuppressive functions. Interestingly, regarding these M2 markers, a better efficiency of murlentamab compared to the normal fucosylated form 3C23K-CHO was demonstrated (Figure 3A). Moreover, although the opsonization of tumor cells with murlentamab did not modify the proportion of TAMs expressing CD163, it decreased the proportion of TAMs positive for CD36 and CD206 (Figure 3A).

Consistently with the increased proportion of MDMs expressing M1 markers, we observed that the co-culture of murlentamab-opsonized SKOV3-R2^+^ tumor cells with both M0 and TAM-like MDMs induced a significant increase in IL1β, IL6 and IFNγ pro-inflammatory cytokine production (Figure 3B). Moreover, IL12 was only induced by the the co-culture of murlentamab-opsonized SKOV3-R2^+^ tumor cells with M0 while TNFα secretion was not modified (Figure 3B). Inversely, the production of the IL10 anti-inflammatory cytokine was significantly reduced by the co-culture of M0 MDMs with SKOV3-R2^+^ cells opsonized with murlentamab and 3C23K-CHO with a better efficiency of the murlentamab low-fucosylated form, whereas no effects were demonstrated on TAM-like MDMs (Figure 3B).

Regarding pro-inflammatory chemokines involved in the attraction of innate and adaptive immune cells, the production of CCL2 was increased by the co-culture of TAM-like MDMs with SKOV3-R2^+^ cells opsonized with murlentamab (Figure 3B). Moreover, we observed that the production of CCL4, CCL5, CXCL9 and CXCL10 chemokines was induced by the co-culture of murlentamab-opsonized SKOV3-R2^+^ tumor cells with both M0 and TAM-like MDMs, while the normal fucosylated form 3C23K-CHO did not demonstrate any effects (Figure 3B).

In order to unequivocally establish that changes in secreted cytokines observed in the culture medium were linked to a modification of the secretory profile of macrophages rather than tumor cells, TAMs were cultured without any tumor cells in the presence of immobilized antibodies (Appendix A). In line with previous results, the culture of TAMs with the murlentamab antibody induced a significant increase in TNFα and IL6 pro-inflammatory cytokines as well as a decrease in IL10 anti-inflammatory cytokine production (Appendix A).

Altogether, these data demonstrate that the murlentamab opsonization of SKOV3-R2^+^ ovarian tumor cells is able to orient naïve macrophages and, even more, to reprogram the phenotype of TAMs towards an M1-like profile. Moreover, this reprogrammation seems more effective in the presence of the low-fucose murlentamab compared to its normal fucosylated form (3C23K-CHO). Interestingly, these results showing a strong increased production of several immunostimulatory mediators (i.e., IL-12, IFN IL6, IFNγ, CXCL9 and CXCL10), also suggest that murlentamab could promote the ability of macrophages to recruit and/or activate the adaptive immune response.

### 2.4. Murlentamab Promotes the Ability of Naïve and Tumor-Associated Macrophages to Activate the Anti-Tumor Adaptive Immune Response

Regarding our results showing that the murlentamab opsonization of SKOV3-R2^+^ ovarian tumor cells promote the orientation of macrophages towards an M1-like profile expressing soluble mediators involved in T cell recruitment and/or activation (Figure 3), we explored its impact on the adaptive immune cells, especially CD4^+^, CD8^+^ and Tregs. For this purpose, autologous human T cells were added in the co-culture with human unstimulated or TAM-like MDMs and SKOV3-R2^+^ tumor cells opsonized with the different antibodies previously described (murlentamab, 3C23K-FcKO control or 3C23K-CHO). At day 10, after 4 days of co-culture, the phenotypic profile of CD4^+^, the activation of CD8^+^, as well as the proportion of Tregs were evaluated using flow cytometry (Appendix A).

As a first step, we investigated whether the co-culture with T cells altered the ability of human MDMs to release pro-inflammatory cytokines and chemokines (Appendix A). While the addition of T cells into co-culture promoted the ability of TAM-like MDMs cultivated with murlentamab-opsonized SKOV3-R2^+^ to release IFNγ, IL6 and IL23, it decreased their ability to produce IL10 (Appendix A). Moreover, IL12 and IFNγ secretions were induced in M0 MDMs cultivated with murlentamab-opsonized SKOV3-R2^+^. We also observed that the addition of T cells into co-culture induced the ability of both M0 and TAM-like MDMs cultivated with murlentamab-opsonized SKOV3-R2^+^ to release high amounts of CCL4, CCL5, CXCL9 and CXCL10 pro-inflammatory mediators (Appendix A). This cytokinic signature in favor of a strong immunostimulatory microenvironment suggests a possible murlentamab-dependent increase in the effector T cells (Th1 and Th17) versus Tregs.

Regarding the Th1/Th2 profile of CD4^+^ T cells, the co-culture of TAM-like MDMs with SKOV3-R2^+^ cells opsonized with murlentamab and 3C23K-CHO enhanced the proportion of Th1 CD4^+^ T cells while lowering the proportion of Th2 CD4^+^ T cells (Figure 4A). Moreover, the same tendency was observed when murlentamab- and 3C23K-CHO-opsonized SKOV3-R2^+^ tumor cells were co-cultured with M0 MDMs (Figure 4A). Remarkably, we demonstrated that the co-culture of both M0 and TAM-like MDMs with SKOV-R2^+^ cells opsonized with murlentamab and 3C23K-CHO resulted in a strong reduction of the proportion of Tregs (Figure 4B). Furthermore, in the presence of M0 and TAM-like MDMs, SKOV3-R2^+^ cells opsonized with 3C23K-CHO tended to increase the proportion of CD8^+^ CD183^+^ T cells, and the opsonization of tumor cells with murlentamab significantly increased activated CD8^+^ T cell proportion (Figure 4C).

Interestingly, these findings were extended to another human ovarian tumor cell line modified to overexpress AMHRII (COV434-R2^+^) (Appendix A). Indeed, as observed with SKOV3-R2^+^ cells, we showed an orientation of CD4^+^ T cells towards a Th1 profile as well as an activation of CD8^+^ T cells when murlentamab-opsonized COV434-R2^+^ were cultured with TAM-like MDMs (Appendix A). Altogether, these data indicate that the opsonization of ovarian tumor cells with murlentamab promotes the activation of an effective anti-tumor T cell immune response.

### 2.5. Murlentamab and Pembrolizumab Association Enhances the Anti-Tumor Potential of Murlentamab Monotherapy

Given the increasing success of immunotherapies, in particular anti-PD-1/PD-L1 antibodies, we assessed both in vitro and in vivo the impact of the association between murlentamab and pembrolizumab, an anti PD-1 antibody, on tumor cell elimination and immune cell activation.

The addition of pembrolizumab in the co-culture between TAM-like MDMs and SKOV3-R2^+^ cells did not change the kinetic of tumor cell number observed with 3C23K-FcKO-opsonized SKOV3-R2^+^ (Figure 5A). Interestingly, while the number of murlentamab-opsonized SKOV3-R2^+^ cells was significantly reduced after 2 days of co-culture with TAM-like MDMs as compared to the 3C23K-FcKO-opsonized SKOV3-R2^+^ cells, the addition of pembrolizumab into the culture medium enhanced the elimination of murlentamab-opsonized SKOV3-R2^+^ cells by decreasing their number from the first day of co-culture (Figure 5A).

As already observed in Figure 4, the opsonization of ovarian tumor cells with murlentamab, in the presence of TAM-like MDMs, promoted the activation of both CD4^+^ T cells towards a Th1 profile and CD8^+^ T cells (Figure 5B,C). Moreover, the addition of pembrolizumab to 3C23K-FcKO-opsonized SKOV3-R2^+^ similarly promoted Th1 orientation of CD4^+^ T cells and CD8^+^ T cell activation. Interestingly, the stimulation of this anti-tumor adaptive immune response was improved by the addition of pembrolizumab to the murlentamab-opsonized SKOV3-R2^+^ cells co-cultured with TAM-like MDMs (Figure 5B,C), demonstrating a potentiated effect during murlentamab/pembrolizumab combo-therapy.

In order to confirm our in vitro observations, the efficacy of the murlentamab/pembrolizumab combination therapy was then evaluated in vivo using xenografted COV434-R2^+^ cells, after verification of PD-L1 expression at their surface (data not shown). The assay used was particularly delicate because murlentamab does not cross-react with murine AMHRII. It was therefore necessary to transplant COV434-R2^+^ cells into immunodeficient NOG (NOD/Shi-scid/IL2Rγ^null^) mice and to re-constitute immune response with an intravenous. injection of human CD34^+^ cells. Moreover, to stabilize monocyte population for several weeks in such a model, mice were treated with a GM-CSF/IL3/IL4 combo. This treatment maintained monocytes during all the experiment but, in parallel, favored GvH reaction. For this reason, the experiment did not exceed 4 weeks and a clear-cut anti-tumoral effect related to a murlentamab-dependent immunomodulation was difficult to demonstrate. However, even with these limits, we observed that the treatment of humanized ovarian tumor-bearing mice by murlentamab monotherapy tended to increase CD86^+^ blood cells and strongly decreased CD163^+^ blood cells (Figure 5D), signing a monocyte/macrophage orientation towards an M1 anti-tumor profile. Although the pembrolizumab monotherapy did not impact the proportion of blood cells expressing CD86 and CD163, the murlentamab/pembrolizumab association maintained and even more enhanced this M1-like profile (Figure 5D). Consistent with these findings, we observed a tendency towards tumor growth inhibition in mice treated with murlentamab or pembrolizumab monotherapies (Figure 5E). In addition, although not significant (murlentamab vs. combo at D24: *p* = 0.101 as determined with the Mann–Whitney one tail test), we showed a trend towards a potentiated effect with the murlentamab/pembrolizumab combo-therapy (Figure 5E). Altogether, these in vitro and in vivo results show a rationale for combining murlentamab with anti-cancer immunomodulators such as pembrolizumab.

## 3. Discussion

Monoclonal antibodies are the fastest growing class of biological therapeutics for the treatment of various cancers. Their strong anti-tumor potential mainly relies on the Fc-mediated immune effector functions, namely, ADCC and ADCP. Reduced core fucosylation of antibodies has been shown to increase IgG1 Fc binding affinity to the CD16a present on immune effector cells (especially NK cells and macrophages) [16], to increase ADCC/ADCP [17,18] and to enhance tumor inhibition in vivo [43,44]. These findings led to the production of low-fucose glycol-engineered antibodies for cancer therapy, a family to which murlentamab belongs.

Murlentamab is a humanized low-fucose anti-AMHRII antibody currently under a phase I and a phase IIa clinical evaluation with preliminary data showing anti-tumor activity through a partial response and prolonged PFS [45,46]. In terms of mode of action, preclinical studies of murlentamab demonstrated its high efficiency for binding to the CD16a receptor and for inducing ADCC/ADCP [13,14]. In vivo, this activity was translated into anti-tumor activity against Patient Derived Xenograft (PDX) models and xenografted cells expressing AMHRII [13,14]. An ex vivo study has also suggested that murlentamab anti-tumor mechanisms could be also related to the increase in macrophage-dependent ADCC/ADCP following CD16a recognition as well as the alleviation of T cell immunosuppression by these cells [14]. Recently, Prat and colleagues have demonstrated changes in blood monocyte subsets after murlentamab infusion in patients with ovarian cancers [38], thus reinforcing the idea that murlentamab activity could involve monocytes/macrophages.

Following these results, we further investigated the precise in vitro mechanisms involved in this antibody anti-tumor activity with a particular focus on macrophage polarization. Indeed, macrophages can have a dual inhibitory and supportive influence on cancer depending on the disease stage, the tissue involved, the host microbiota and their polarization state whose two extremes are described as M1 and M2 macrophages [47]. Moreover, macrophages play a pivotal role within the tumor microenvironment since, in addition to displaying direct pro- or anti-tumoral activities, they orchestrate the adaptive immune response [47]. In this work, through an in vitro co-culture system allowing the interactions between the different partners involved in the anti-tumor immune response (namely, antibody-opsonized tumor cells, naïve or tumor-educated human MDMs and naïve T cells), we accurately dissected the murlentamab mode of action.

First, we demonstrated that murlentamab treatment resulted in several functional and phenotypic modifications of naïve MDMs (M0). Indeed, as expected and in line with previously cited studies, recognition of murlentamab-opsonized ovarian tumor cells led to a strong ADCC activity of naïve macrophages. Moreover, the murlentamab opsonization of tumor cells induced a phenotypic orientation of M0 macrophages toward an M1-like phenotype. Indeed, in the presence of murlentamab-opsonized tumor cells, macrophages adopt a pro-inflammatory profile (secreting IL12, TNFα, IL6, IL1β, IFNγ, CXCL9 and CXCL10) with the overexpression of surface receptors characteristic of the M1 polarization (CD80, TLR2) while downregulating several M2-like markers (CD163, CD206, CD36 and IL10). Other studies have already demonstrated that FcγR-mediated signaling has a profound impact on unstimulated monocyte/macrophage. Indeed, engagement of activating FcγRs on these cells is associated with the production of pro-inflammatory cytokines and chemokines including IL-8, TNF and IL1β [48,49]. In our study, the observed phenotypic switch, although visible with the non-glycoengineered version of the antibody (3C23K-CHO), was potentiated in the presence of the low-fucose antibody (murlentamab). In line with these results showing a glycosylation profile-related effect, Kircheis and colleagues demonstrated that the pro-inflammatory secretion profile of peripheral blood mononuclear cells (PBMCs) was particularly induced using glyco-modified antibody compared to their parental counterpart [49]. Thus, our results confirm that glyco-engineered antibodies not only present a stronger binding to CD16a and a higher lysis potency through ADCC/ADCP but also a greater ability to promote a pro-inflammatory cytokine release.

Moreover, in an original way, we demonstrated here that murlentamab is not only capable of orienting naïve human MDMs towards an M1-like phenotype but also of reprogramming polarized TAMs. Indeed, as previously described with M0 macrophages, the co-culture with murlentamab-opsonized ovarian tumor cells results in (i) an increased TAM ADCC and (ii) a repolarization of TAMs towards an M1 profile secreting a high amount of pro-inflammatory and immunostimulatory cytokines/chemokines. Knowing that tumor cells constantly secrete immunosuppressive factors to educate immune cells [29], these results suggest a promising anti-tumor role of murlentamab in clinical settings, more specifically in cancer patients in whom TAMs represent the major leukocyte infiltrate within a tumor.

In addition to the phenotypic switch of macrophages and the activation of their direct anti-tumor properties, we demonstrated here that murlentamab stimulated an adaptive anti-tumor immunity. Indeed, in co-culture with macrophages and murlentamab-opsonized tumor cells, we reported an increased proportion of activated CD8^+^ and Th1 CD4^+^ T cells as well as a reduced proportion of CD25^+^ CD4^+^ Tregs. This is in line with the pro-inflammatory profile of macrophages which suggest a macrophage-dependent T cell activation after opsonized tumor cell recognition and FcγR engagement. Indeed, IL12 and IFNγ cytokines can affect immune function in several ways, for instance, through (i) the enhancement of cross-presentation by professional antigen-presenting cells, (ii) the augmentation of co-stimulatory molecule expression, including major histocompatibility complex (MHC) I and MHCII, (iii) the polarization of T cells into Th1 effector cell phenotype and (iv) the stimulation of cytotoxic CD8^+^ T cell proliferation [50,51]. Opening up promising therapeutic perspectives, we also transposed the results obtained in our in vitro model to the clinic. Indeed, through the analysis of two clinical studies carried out in patients with gynecological and advanced/metastatic colorectal cancers, we demonstrated that murlentamab treatment is associated with several signs of innate and adaptive immune cell activation both systemically and locally at the tumor site. Altogether, our results showing an increased CD16/Granzyme B-positive cell infiltration after murlentamab infusion and very few NK cells at the tumor site (in all biopsies tested, less than 0.1% of cells were positive with a staining using an anti-NKp46 antibody) suggest a monocyte/macrophage-dependent anti-tumor mechanism. Supporting this hypothesis, Uchida and colleagues demonstrated that monocytes and/or macrophages and not NK cells are the principal mediators of ADCC against α-CD20-coated B cells in vivo [52]. Regarding solid tumors, participation of macrophages in mediating anti-tumor mAb efficacy is less established but at least one report is consistent with this notion [18]. Interestingly, this work further emphasizes the enhanced efficacy of glyco-engineered antibodies (enhanced CD16a binding activity) compared to their wild-type counterpart [18]. Moreover, the decreased proportion of monocytes positive for the CD163, a well-characterized marker of TAMs, supports the murlentamab-dependent modulation of monocytes/macrophage populations/phenotypes. Of interest, CD163 expressing TAMs have been linked to poor prognosis, overall survival and metastasis of a range of malignancies, including colorectal and ovarian cancers [39]. Finally, the two clinical studies show some evidence of T cell activation in patient’s blood and tumor biopsies after murlentamab infusion. In this framework, it has already been demonstrated that the mAb-mediated induction of IFNγ resulted in DC maturation and increased antigen presentation, which was hypothesized to result in increased cross-presentation potential to CD8^+^ T cells, thereby linking the innate and adaptive immune responses [53]. In addition, the elevation in patients’ blood of CXCL9 and CXCL10 chemokines, the levels of which are increased in the co-culture medium after murlentamab-opsonized tumor cell addition, could also partly explain the murlentamab anti-tumor activity by promoting the recruitment of activated CD8^+^ and Th1 lymphocytes.

Therapeutic strategies targeting macrophages and aiming to reactivate or re-educate them are currently undergoing clinical assessment [47]. Such strategies have the potential to complement cytoreductive, antiangiogenic, and immune-checkpoint-inhibitor treatments. Interestingly, through in vitro assays and delicate in vivo experiments, our work underscores the improved effect of murlentamab/pembrolizumab bi-therapy compared to murlentamab used as a monotherapy. Moreover, we confirmed in vivo some signs of murlentamab-induced macrophage phenotypic orientation towards an M1-like profile. This polarization shift could contribute to the enhanced effect observed here with pembrolizumab. Although preliminary, these data support the interest of carrying out additional in vivo experiments in order to (i) determine the optimal treatment sequence with these two agents, (ii) compare the activity of murlentamab versus its normal fucosylated form as widely explored in our in vitro studies and (iii) further investigate immune cell activation in both mice blood samples and xenografted tumors. However, this first in vivo experiment already confirmed that murlentamab anti-tumor activity is not only restricted to its ability to promote ADCC/ADCP but also involves a more global activation of the immune system. In this framework, a recently published study highlighted the fundamental role of macrophage-derived CXCR3 ligands for the therapeutic efficacy of immune checkpoint blockade, thus highlighting the potential of manipulating this axis to enhance patient responses [54]. Given our results showing a murlentamab-dependent induction of CXCL9 and CXCL10 macrophage secretion, this stresses the importance of our study and supports the benefits of associating murlentamab with other types of immunotherapies such as antibodies against PD1, PDL1, CTLA4, or CD47.

## 4. Materials and Methods

### 4.1. Cell Lines and Reagents

Cells from the human germ cell tumor cell line COV434 [55] were transfected with the cDNA encoding full-length human AMHRII in the pCMV6 plasmid to stably express AMHRII and to constitute the COV434-R2^+^ cell line, as described by Kersual et al. [8]. Cells were grown in DMEM (Dubecco’s Modified Eagle Medium) F12 medium containing 10% heat-inactivated fetal bovine serum, 0.1 mg/mL streptomycin, 0.1 IU/mL penicillin and 0.25 µg/mL amphotericin B. COV434-R2^+^ cells were supplemented with 0.33 mg/mL geneticin. Cells from the human ovarian adenocarcinoma cell line SKOV3 were also transfected with the cDNA encoding full-length human AMHRII in the pCMV6 plasmid as described with COV434 cells. SKOV3 cells were maintained in DMEM medium supplemented with 10% heat-inactivated fetal calf serum, 100 U/mL penicillin and 100 μg/mL streptomycin. All cells were grown at 37 °C in a humidified atmosphere with 5% CO_2_ and the medium was replaced twice a week. Cells were harvested with 0.5 g/mL trypsin/0.2 mg/mL EDTA. All culture media and supplements were purchased from Gibco, Thermo Fisher scientific (Waltham, MA, USA).

### 4.2. Patients and Study Design

The primary analyses derive from two clinical studies: (i) the C101 study, a dose-ranging phase I study of murlentamab single agent or in combination with carboplatin-paclitaxel and its expansion with murlentamab single agent in patients with previously treated advanced or metastatic gynecological cancer (NCT02978755; [45]), and (ii) the C201 study, a phase IIa study of murlentamab single agent or in combination with Lonsurf^®^ (trifluridine/tipiracil) in patients with previously treated advanced or metastatic colorectal cancer (NCT01668784; [46]). In the C101 study (Appendix A), murlentamab was administered to the patients either every two weeks (q2w): on days 1 (D1) and 15 (D15) of each 4-week cycle, or, as in the C201 study (Appendix A), every week (q1w): on days 1, 8, 15 and 22 (D1, D8, D15, D22) of each 4-week cycle. Details on study treatment and schedule are provided in Appendix A.

### 4.3. Tumor and Blood Specimens

All human tissues and bloods were obtained according to protocols approved by institutional review boards from each country where the studies where performed (Comité de protections des personnes, Hôpital Tarnier-Cochin, Paris, France; Comité d’éthique, Institut Jules Bordet, Brussels, Belgium; Ethics committee research UZ/KU, Leuven, Belgium; London Chelsea, London, UK; Ethics committee for multicentric clinical trial, Motol hospital, Prague, Czech Republic). For both studies, C101 and C201, biopsies and blood samples were obtained from each patient who provided voluntary written informed consent. Circulating immune cells were analyzed at the first day of the first cycle before murlentamab infusion (baseline, D1) and, at day 15 of cycle 1 (D15) and at day 1 of cycle 3 (D43). In the phase I (C101) study, these cells were analyzed with a panel of specific markers for each investigational site, whilst, in the phase IIa (C201) study, all samples from Belgian investigational sites were centralized at Jules Bordet Institute, Brussels, for analysis. In both studies, few tumor specimens have been collected before murlentamab infusion (baseline) and at the end of cycle 2 (D22). Biopsies were fixed in 10% formalin solution, processed, and embedded in paraffin.

### 4.4. Phenotype Analysis of Blood Circulating Immune Cells of Patients

In the C201 study, CD69^+^ cells were detected by incubating 100 µL of fresh total blood with 20 µL of anti-human Fc receptor binding inhibitor (ThermoFisher, Bleiswijk, The Netherlands) for 20 min on ice and then with fluorescent antibodies for 20 min at 4 °C (CD3-VioBlue, CD14-APC-vio770, CD45-VioGreen from Miltenyi Biotec and CD4-Alexa Fluor700, CD15-eF450, CD56-FITC, CD64-PerF710, CD16-Pc7, CD25-PercPeF710, CD69-PE, CD127-APC from Thermofisher, Bleiswijk, The Netherlands) according to the manufacturer’s suggested dilutions. After incubation, 1 mL of diluted Red Blood Cell lysis buffer (Miltenyi, Bergisch Gladbach, Germany) was directly added and cells were incubated for 10 min at room temperature in the dark. After centrifugation, cells were immediately acquired on a GALLIOS 10/3 cytometer (Beckman Coulter, Nyon, Switzerland) and results were analyzed with the Kaluza Flow Cytometry Analysis v1.5 software. CXCL9 and CXCL10 quantification was performed using, respectively, Quantikine^®^ ELISA Human CXCL9/MIG Immunoassay and Quantikine^®^ ELISA Human CXCL10/IP-10 Immunoassay kits (R & D System, Abingdon, UK) according to manufacturer’s protocols.

In the C101 study, the rates of CD8^+^ T cell activation were determined by ICOS expression according to a similar protocol but using another panel of fluorescent antibodies (CD3-AA700, CD4-PB, CD8-KrO, CD25-PC5.5, CD127-AA750, CD45RA-PC7, CD197-PE, CD95-FITC, CD278-APC, HLA-DR-ECD from Beckman Coulter, Marseille, France). The rates of naïve and memory T cells, Tregs and T cell activation were determined by HLA-DR and ICOS expression according to a similar protocol but using another panel of fluorescent antibodies (CD3-AA700, CD4-PB, CD8-KrO, CD25-PC5.5, CD127-AA750, CD45RA-PC7, CD197-PE, CD95-FITC, CD278-APC, HLA-DR-ECD from Beckman Coulter, Marseille, France). Moreover, detection of circulating CD163^+^ monocytes was determined by using CD14-PE and CD163-Vioblue (Myltenyi Biotec, Bergisch Gladbach, Germany).

### 4.5. Immunofluorescence

The multiplex immunofluorescence assay was performed on the Ventana Discovery ULTRA automated slides Stainer (Ventana, Tucson, AZ, USA). This technology uses sequential application of unmodified primary antibodies with, among each, a specific heat deactivation maintaining tissue and antigen epitope integrity and avoiding cross reactivity between reagents. After dewax and pretreatment, FFPE (Formalin-Fixed Paraffin-Embedded) slides were incubated with primary antibody CD16 (clone SP175, Ventana Medical Systems, Mannheim, Germany), CD8 (clone C8/144B, Agilent, DAKO, Glostrup, Danemark), Granzyme B (clone GRB7, Agilent, DAKO, Glostrup, Danemark), ICOS (clone D1K2T, Cell Signaling Technologies, Leiden, The Netherlands), CD86 (clone E2/G8P, Cell Signalling Technologies, Leiden, The Netherlands) CD163 (clone E2/G8P, Cell Signalling Technologies, Leiden, The Netherlands) and CD14 (clone EPR3653, Roche Diagnostic, Mannheim, Germany). Primary antibodies were visualized using the OmniMap-HRP (Horse radish peroxidase conjugated anti-rabbit, anti-mouse), Amplification anti-HQ HRP Multimer secondary systems and tyramide-conjugated fluorophore kits FAM, Red610, Rhodamin6G and Cy5 (Ventana, Tucson, AZ, USA). Counterstain was performed using Hematoxylin and Bluid Reagent (Ventana, Tucson, AZ, USA). Multiplex Immunofluorescence stained slides were scanned using a P250 Panoramic digital scanner from 3D Histech (Budapest, Hungary) with appropriate fluorophores used (Rhodamin6G, RED610, FAM, Cy5). Immunofluorescence image quantifications were performed using indicaLabs HALO Imaging Analysis software (PerkinElmer, Villepinte, France).

### 4.6. Preparation of Human Monocyte-Derived Macrophages (MDMs)

Peripheral blood mononuclear cells (PBMCs) were obtained from healthy blood donors (Etablissement Français du Sang, EFS, France). Written informed consents were obtained from the donors under EFS contract number 21/PVNT/TOU/IPBS01/2009-0052. According to articles L1243-4 and R1243-61 of the French Public Health Code, the contract was approved by the French Ministry of Science and Technology (agreement number AC 2009-921) (Paris, France).

Human Monocytes were isolated from PBMCs using negative selection Monocyte Isolation Kit II (Macs Miltenyi), as recommended by the manufacturer’s protocol. Monocytes were cultured at 37 °C and 5% CO_2_ in Macrophage-Serum Free Media (SFM, Gibco, Scotland, UK) supplemented with l-glutamine (Invitrogen, Bleiswijk, The Netherlands) and penicillin/streptomycin (PS, Invitrogen, Bleiswijk, The Netherlands). Isolated monocytes were differentiated to pro-tumoral macrophages over three days by M-CSF (Macs Miltenyi, 200 UI/mL) and IL-10 (Macs Miltenyi, 50 UI/mL) stimulation.

### 4.7. Opsonization and Co-Culture of Ovarian Carcinoma Tumor Cell Lines with Human MDMs

Human ovarian cancer cell lines were labeled with the murlentamab anti-AMHRII antibody (3C23K or GM102^®^, 10 µg/mL), 3C23K-FcKO (an isotypic control, 10 µg/mL) or 3C23K-CHO (a normal fucose form, 10 µg/mL) at 4 °C during 1 h. Opsonized tumor cells were then resuspended in Dulbecco’s modified Eagle’s medium (DMEM, Gibco, Scotland, UK), supplemented with l-glutamine, PS and 10% heat-inactivated fetal calf serum (FCS, Sigma, Salisbuty, UK) and added to the differentiated human macrophages at a 1:1 ratio.

In some experiments, anti-AMHRII antibodies were used in combination with an anti-PD-1 antibody, pembrolizumab (Selleck Chemicals, Selleck, USA) at 10 µg/mL. This antibody was added every day in the co-culture until T cells cytometry analysis.

### 4.8. Evaluation of Ovarian Carcinoma Tumor Cell Number

SKOV3-R2^+^ or COV434-R2^+^ cells were stained with the CellTrace^TM^ Violet Cell Proliferation kit (Molecular Probes™, Life technology, Bleiswijk, The Netherlands), opsonized with anti-AMHRII antibodies (3C23K, 3C23K-FcKO or 3C23K-CHO) and added to cultures of differentiated human macrophages at a 1:1 ratio. At different timepoint of the co-culture (Days 3, 4 and 5 according to the experimental protocol represented in Appendix A), SKOV3-R2^+^ cell number was evaluated by detecting fluorescently labeled cells by flow cytometry.

### 4.9. Antibody-Dependent Cell-Mediated Cytotoxicity (ADCC) Assay

After opsonization with the different anti-AMHRII antibodies, target SKOV3-R2^+^ cells were loaded with BATDA (Bis-acetoxymethyl-2,2′:6′,2″-terpyridine-6,6″-dicarboxylate), resuspended in DMEM (Gibco, Scotland, UK) supplemented with l-glutamine, PS, and 10% heat-inactivated FCS and added tothe effector cells (human macrophages) at a 1:1 ratio, at 37 °C for 4 h.

ADCC was measured by using the DELFIA EuTDA-based cytotoxicity assay (PerkinElmer, Waltham, MA, USA). After 4 h of the co-incubation of target and effector cells, supernatants were then incubated with Eu^3+^ solution and fluorescence was measured (Envision, PerkinElmer, Waltham, MA, USA). Data were normalized to maximal (target cells with triton) and minimal (effector cells alone) lysis and fit to a sigmoidal dose–response model.

### 4.10. Phenotypic Characterization of Human MDMs

Receptor expression (CD32, CD64, CD80, TLR2, CD163, CD36 and CD206) was evaluated by flow cytometry at the membrane of differentiated human macrophages after 3 days of co-culture with opsonized SKOV3-R2^+^ tumor cells. Receptors were detected using CD32-PE-Vio770, CD64-PerCP-Vio700, CD80-PE, CD282 (TLR2)-APC, CD163-PE, CD36-PE and CD206-APC (Miltenyi, Bergisch Gladbach, Germany) and were compared with an appropriate isotype control.

A population of 10,000 cells was analyzed for each data point. Dead cells (positive cells) were removed from the analysis after labeling with Viobility Fixable Dye (Miltenyi, Bergisch Gladbach, Germany). Analyses were gated on CD14 or Cd11b positive cells. All analyses were performed using a BD Fortessa flow cytometer with the Diva software. The gating strategies are presented in the Appendix A.

### 4.11. Production of Cytokines and Chemokines

Cytokines (IL1β, IL12, TNFα, IL6, IFNγ, IL23 and IL10) and chemokines (CCL2, CCL4, CCL5, CXCL9 and CXCL10) release was quantified in the supernatant after 3 days of co-culture between differentiated human macrophages and opsonized SKOV3-R2^+^ tumor cells and 4 days after T cell addition according to the manufacturer’s instructions (AlphaLISA, PerkinElmer, Waltham, MA, USA).

### 4.12. Human T Cell Isolation

Human T cells were isolated from PBMCs using negative selection Pan T Cell Isolation Kit (Macs Miltenyi, Bergisch Gladbach, Germany) and activated (T cell activation/Expansion kit, Macs Miltenyi, Bergisch Gladbach, Germany) as recommended by the manufacturer’s protocol. T cells were resuspended in RPMI 1640 Medium (Gibco, Scotland, UK) supplemented with l-glutamine, PS, and 10% heat-inactivated FCS and added in the co-culture wells with human macrophages and opsonized SKOV3-R2^+^ tumor cells at a 1:8 ratio for 4 days.

### 4.13. T Cell Polarization and Activation

To profile lymphocyte populations, cells were labeled with the following antibodies: CD45-VioGreen, CD3-APCVio770, CD183 (CXCR3)-APC, CD25-PE, CD8-PEVio770 and CD4-ViobrightFITC (Macs Miltenyi, Bergisch Gladbach, Germany). A population of 10,000 cells was analyzed for each data point. All analyses were performed using a BD Fortessa flow cytometer with the Diva software. The gating strategies are presented in the Appendix A.

### 4.14. Secretory Profile of M2-Like Macrophages in the Presence of Murlentamab

M2-like macrophages were generated in vitro by culturing CD14^+^ monocytes, isolated from healthy donors (Etablissement Français du Sang, Toulouse, France; EFS), in the presence of complete culture medium containing 50 ng/mL recombinant human M-CSF for 4 days, as previously described [56]. In parallel, anti-AMHRII antibodies were prepared in sterile and apyrogen PBS (10 mM, pH = 7.4; Lonza, Basel, Switzerland) and adsorbed for 24 h at 4 °C to 24-well cell culture plates (10 µg/mL, 1 mL/well), as previously described. M-CSF-differentiated macrophages were cultured for 3 more days in antibody-coated plates (10^6^ cells/well). Cells were then stimulated or not for 24 h with 100 ng/mL LPS (Sigma-Aldrich, Salisbuty, UK), in the absence of the antibody, because, as previously reported, human macrophages have to be stimulated to reveal their phenotype [29]. The expression of CD16, CD64, and CD163 surface molecules was evaluated with, respectively, anti-CD16 (clone 20G8; BD Biosciences, Erembodegem, Belgium), anti-CD64 (clone 10.1; BD Biosciences, Erembodegem, Belgium) and anti-CD163 (clone 215927; R&D Systems, Abingdon, UK) by flow cytometry using a FACSCanto II flow cytometer (BD Biosciences, Erembodegem, Belgium) and analyzed with the FlowJo software. IL1β, 6, TNFα, IL10 and IL12 in the culture supernatants of 24 h stimulated cells were quantified by ELISA (Diaclone, Besançon, France).

### 4.15. In Vivo Experiment

All procedures were reviewed and approved by the local ethics committee (Comité d’Ethique pour l’Expérimentation Animale Genevois, ethic code: A740169; CELEAG). The experiments were carried out with four-week-old NOG (NOD/Shi-scid/IL-2Rγnull) immunodeficient mice^®^ (Taconic, Doussard, France) engrafted with cord blood-derived CD34^+^ hematopoietic stem and progenitor cells (French Blood Bank, Annemasse, France) two days after chemical myeloablative treatment. Engraftment consisted in intravenous injection of CD34^+^ cells. Fourteen weeks after cell injection, engraftment level was monitored with the analysis of human CD45^+^ cells among total blood mouse and human leukocytes by flow cytometry (Attune, Gibco, Thermo Fisher scientific, Waltham, MA, USA). At the thirteenth week, macrophage/monocyte cell populations were enhanced in humanized NOG mice by receiving a treatment based on the transient expression of human cytokines GM-CSF/IL-3/IL4. Based on human CD45^+^ analysis, mice were randomized for tumor grafting.

Tumor cells, COV434-R2^+^, were cultured as described previously. After washing steps, cells were suspended in PBS-EDTA (1mM), Matrigel (1:1) and 100 µL of cell suspension (10 × 10^6^ cells) were injected subcutaneously in the right flank of mice. Day 0 was defined as the day of the cell inoculation. The presence of PD-L1 in cells was confirmed before engraftment by flow cytometry (data not shown).

At day 35, when tumors reached an average volume of 60–80 mm^3^, animals were randomized into 4 therapeutic groups of 8 mice based on their humanization rate, their number of myeloid cells (CD11b-positive cells) and their tumor volume. Murlentamab was diluted in PBS then administrated intraperitoneally, twice a week, for 4 weeks. The control groups were treated with PBS (vehicle) with the same schedule of administration as murlentamab. Pembrolizumab (MelonePharma, Dalian, China) was administrated per os at 50 mg/kg in water with 5% dextrose every day for 4 weeks. The fourth group was treated with a combo of pembrolizumab and murlentamab administrated as described above for monotherapy. Tumor growth was evaluated by measuring with a caliper two perpendicular tumor diameters twice a week. Tumor volume and tumor growth inhibition (TGI) were calculated according to the standard method [57]. In vivo toxicity was assessed as follows: weights of individual mice were measured every three days. Variation of mice weight as compared to their initial weight and means per group were calculated. A treatment was considered toxic and was stopped once a bodyweight loss of 15% was observed, and animals were sacrificed if body weight loss persisted for three consecutive days. Treatment was also stopped when an individual mouse had a body weight loss of 20% or more.

Some PD markers were followed during in vivo experiments. The blood of each mouse was collected from the retro-orbital sinus on EDTA-coated tubes (Microvette 100 MCVT100-EDTA, Sarstedt, Marnay, France) before tumor engraftment (Day 0) and at sacrifice. A total of 100 µL of whole blood was stained with antibodies and lysed with red blood cell lysis buffer, washed with PBS and suspended in PBS before acquisition on the Attune Nxt Flow cytometer (Gibco, Thermo Fisher scientific, Waltham, MA, USA). In this study, the detection of hCD86 and hCD163 was performed by using, respectively, anti-CD86 Alexa-700 antibody (BD Biosciences, Erembodegem, Belgium) and anti-CD163 APC antibody (Miltenyi, Bergisch Gladbach, Germany).

### 4.16. Flow Cytometry Antibodies

Flow cytometry antibodies used were shown in Table 1.

### 4.17. Statistical Analysis

For each experiment, the data were subjected to one-way ANOVA analysis followed by the means multiple comparison method of Dunnett or Tukey. *p* < 0.05 was considered as the level of statistical significance.

## 5. Conclusions

This study is the first to detail the mode of action of murlentamab, a low-fucose antibody, from its antigen binding to the tumor cell lysis via a cascade of activation of macrophage and lymphocytes in a sole in vitro study/model/schema integrating all the partners. Moreover, signs of activation of innate and adaptative immune response were confirmed in patients treated with murlentamab. These results open new perspectives in terms of combined treatment by associating murlentamab with checkpoint inhibitors or other immuno-modulators as many agents currently under clinical investigation.

## 6. Patents

Patent WO 2018/219956 A1 resulted from the work reported in this manuscript.

## Figures and Tables

**Figure 1 cancers-13-01845-f001:**
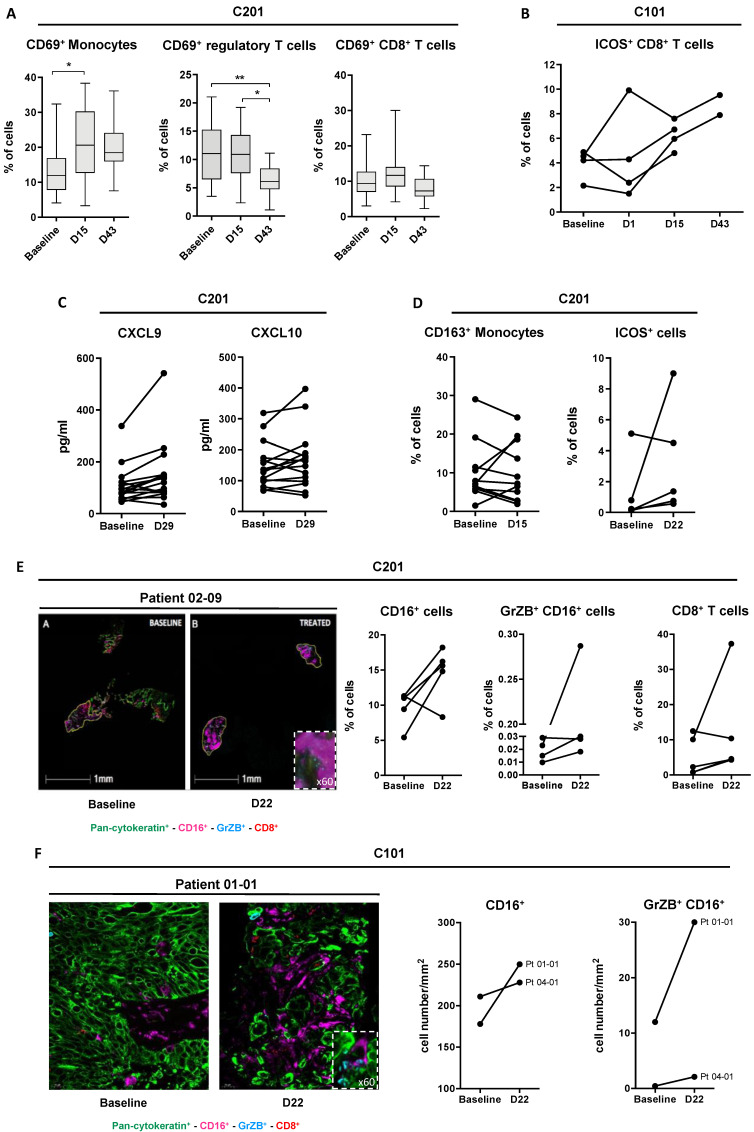
Variation of pharmacodynamic markers measured in blood samples and biopsies of patients from C201 and C101 studies. (**A**) Detection of CD69 as a marker of activation of monocytes, regulatory and CD8^+^ T cells in blood samples of patients (*n* = 20) with colorectal cancer treated with murlentamab (single agent and in combination with trifluridine/tripiracil). Data shown (boxplots) are the results from 20 patients. * *p* < 0.05; ** *p* < 0.01. *p* values were determined using one-way ANOVA analysis followed by Tukey’s multiple comparisons test. (**B**) Detection of ICOS (inducible co-stimulatory molecule) as a marker of lymphocyte activation in blood samples of patients with ovarian cancer treated at Gustave Roussy (Paris, France) with murlentamab in combination with carboplatin + paclitaxel (*n* = 4). (**C**) Detection of CXCL9 (*n* = 16) and CXCL10 (*n* = 15) release in blood samples of all patients treated with murlentamab single agent in C201. (**D**) Detection of co-staining CD14/CD163 (*n* = 8) and of ICOS (*n* = 4) as markers of immune system regulation in FFPE (Formalin-Fixed Paraffin-Embedded) biopsies obtained from the C201 study. (**E**) Image and quantification of CD16 (*n* = 5), co-staining CD16/granzyme B (GrZB) (*n* = 4), and CD8 (*n* = 4) as markers of immune system activation in FFPE biopsies obtained from the C201 study. (**F**) Image and quantification of CD16/granzyme B (GrZB) co-staining in FFPE biopsies obtained from the C101 study (*n* = 2).

**Figure 2 cancers-13-01845-f002:**
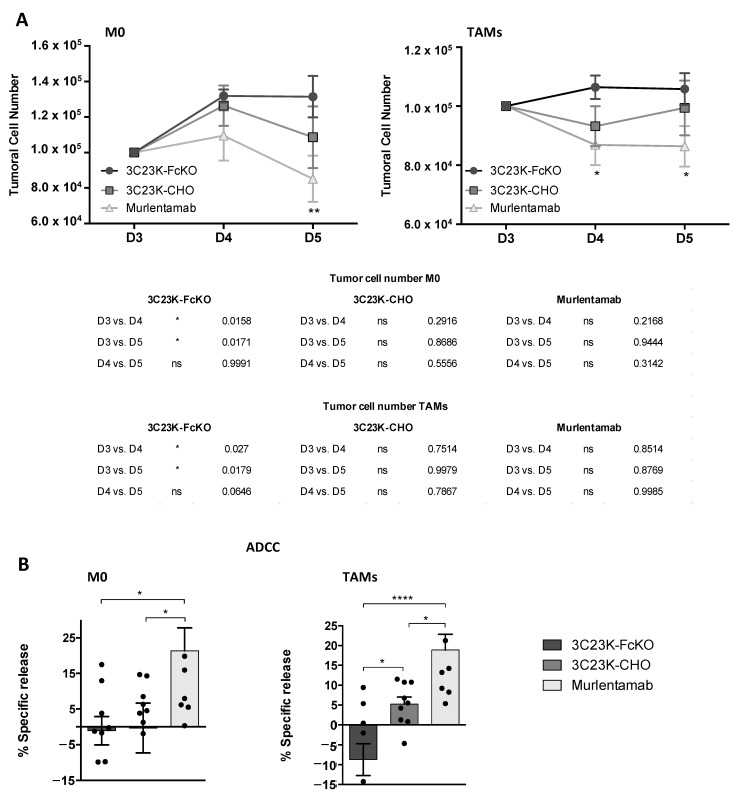
Murlentamab opsonization of SKOV3-R2^+^ increases macrophage anti-tumoral activity and antibody-dependent cell-mediated cytotoxicity (ADCC) killing. SKOV3-R2^+^ ovarian tumor cells were labeled with different 3C23K antibodies (3C23K-FcKO control, 3C23K-CHO normally fucosylated or murlentamab the low fucosylated form) and cultured in the presence of human monocyte-derived macrophages from healthy donors unstimulated (M0) or stimulated with M-CSF and IL-10 (tumor-associated macrophages (TAMs)). (**A**) Opsonized-SKOV3-R2^+^ cell number was determined by flow cytometry after one and two days of co-culture with M0 or TAMs. * *p* < 0.05; ** *p* < 0.01 compared 3C23K-FcKO vs. murlentamab at a given time (**B**) ADCC was performed after 4 h of co-culture between SKOV3-R2^+^ cells and M0 or TAMs. Data shown (mean ± SEM) are the results from three different experiments (performed with three different healthy donors). * *p* < 0.05; **** *p* < 0.0001. *p* values were determined using one-way ANOVA analysis followed by Tukey’s multiple comparisons test.

**Figure 3 cancers-13-01845-f003:**
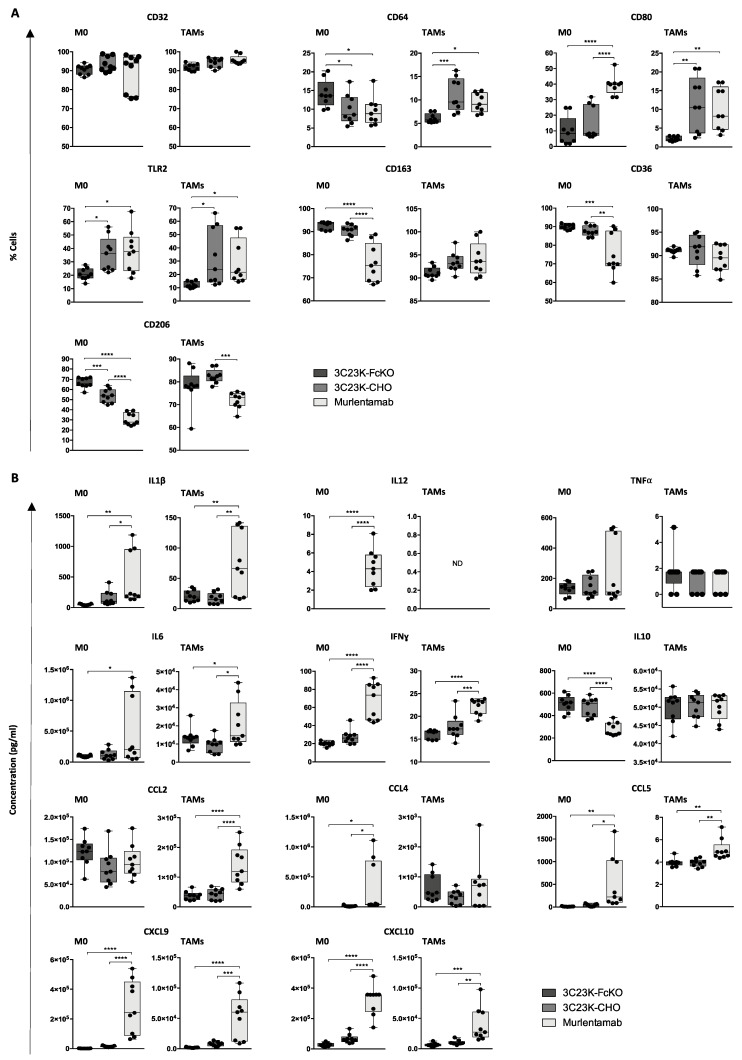
Murlentamab opsonization of SKOV3-R2^+^ orients naïve macrophages and reprograms TAMs towards an M1-like profile. SKOV3-R2^+^ ovarian tumor cells were labeled with different 3C23K antibodies (3C23K-FcKO control, 3C23K-CHO normally fucosylated or murlentamab the low fucosylated form) and cultured in the presence of human monocyte-derived macrophages from healthy donors unstimulated (M0) or stimulated with M-CSF and IL-10 (TAMs). (**A**) The proportion of macrophages expressing M1/M2 membrane markers (CD32, CD64, CD80, TLR2, CD163, CD36 and CD206) was determined by flow cytometry after three days of co-culture with SKOV3-R2^+^ cells. (**B**) The release of cytokines (IL1β, IL12, TNFα, IL6, IFNγ, IL10) and chemokines (CCL2, CCL4, CCL5, CXCL9 and CXCL10) in the culture medium was determined by AlphaLISA after three days of co-culture with SKOV3-R2^+^ cells. Data shown (boxplots) are the results from three different experiments (performed with three different healthy donors). * *p* < 0.05; ** *p* < 0.01; *** *p* < 0.001; **** *p* < 0.0001. *p* values were determined using one-way ANOVA analysis followed by Tukey’s multiple comparisons test.

**Figure 4 cancers-13-01845-f004:**
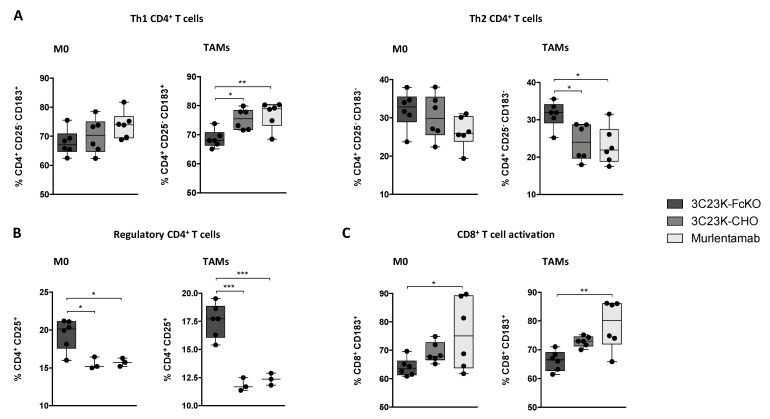
Murlentamab opsonization of SKOV3-R2^+^ activates an effective anti-tumor T cell immune response. SKOV3-R2^+^ ovarian tumor cells were labeled with different 3C23K antibodies (3C23K-FcKO control, 3C23K-CHO normally fucosylated or murlentamab the low fucosylated form) and cultured in the presence of human monocyte-derived macrophages from healthy donors unstimulated (M0) or stimulated with M-CSF and IL-10 (TAMs). After 3 days of co-culture, activated T cells coming from the same healthy donor were added in the culture well for 4 more days. (**A**) The CD4^+^ Th1/Th2 polarization profile, (**B**) the proportion of CD3^+^ CD4^+^ CD25^+^ regulatory T cells and (**C**) the activation of T CD8^+^ cells were determined by flow cytometry after four days of co-culture. Data shown (boxplots) are the results from two different experiments (performed with two different healthy donors). * *p* < 0.05; ** *p* < 0.01; *** *p* < 0.001. *p* values were determined using one-way ANOVA analysis followed by Tukey’s multiple comparisons test.

**Figure 5 cancers-13-01845-f005:**
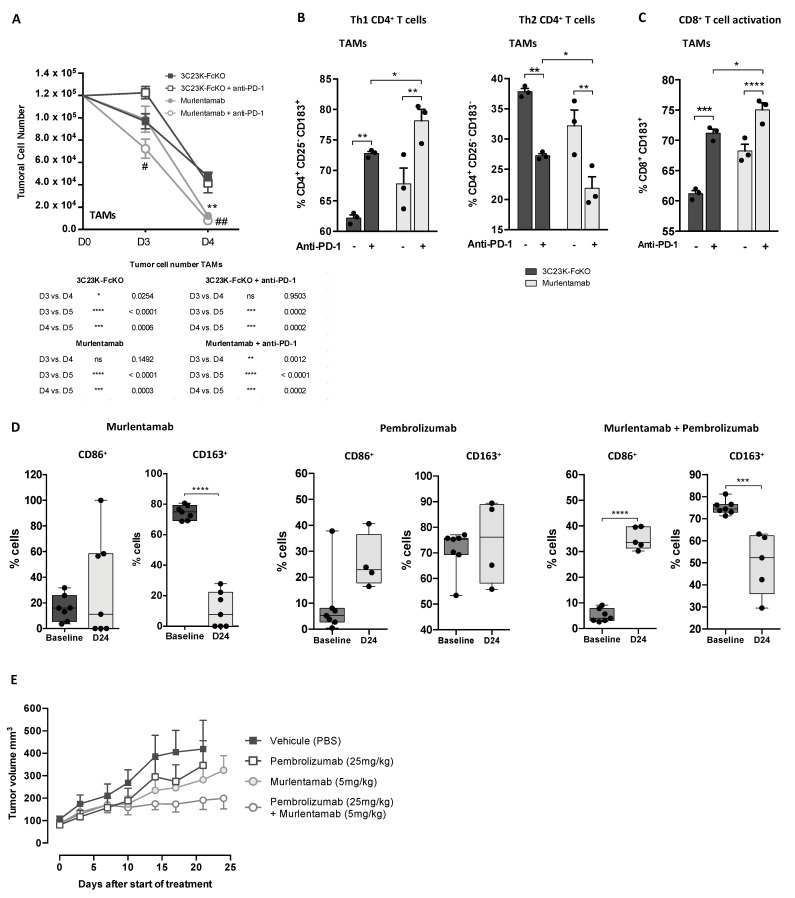
Murlentamab/pembrolizumab combination accentuates the anti-tumoral effect of murlentamab monotherapy through the enhancement of T cell activation. (**A**–**C**) SKOV3-R2^+^ ovarian tumor cells were labeled with different 3C23K antibodies (3C23K-FcKO control or murlentamab the low fucosylated form) and cultured in the presence of human monocyte-derived macrophages from healthy donors stimulated with M-CSF and IL-10 (TAMs). After 3 days of co-culture, activated T cells coming from the same healthy donor were added in the culture well for 4 more days. Pembrolizumab was added into co-culture wells everyday from day 3 to day 10. (**A**) Opsonized-SKOV3-R2^+^ cell number was determined by flow cytometry after one and two days of co-culture with TAMs. Data shown (mean ± SEM) are the results from three different experiments (performed with one healthy donors). ** *p* < 0.01 compared 3C23K-FcKO vs. Murlentamab. # *p* < 0.05; ## *p* < 0.01 compared 3C23K-FcKO + anti-PD-1 vs. Murlentamab + anti-PD-1 as determined using one-way ANOVA analysis followed by Dunnett’s multiple comparisons test. (**B**,**C**) The CD4^+^ Th1/Th2 polarization profile and the activation of T CD8^+^ cells were determined by flow cytometry after four days of co-culture. Data shown (mean ± SEM) are the results from three different experiments (performed with one healthy donors). * *p* < 0.05; ** *p* < 0.01; *** *p* < 0.001; **** *p* < 0.0001. *p* values were determined using one-way ANOVA analysis followed by Tukey’s multiple comparisons test. (**D**,**E**) 10 × 10^6^ COV434-R2^+^ ovarian tumor cells were transplanted subcutaneously into humanized GM-CSF/IL3/IL4 hu-NOG (NOD/Shi-scid/IL2Rγ^null^ ) mice (Taconic). After 35 days, when tumors were big enough, mice were i.p treated or not with murlentamab (5 mg/kg) +/− pembrolizumab (25 mg/kg) twice a week for 4 weeks. (**D**) Quantification of circulating CD86^+^ and CD163^+^ cells by flow cytometry from blood of tumor-bearing mice before treatment and after 24 days of treatment with murlentamab (5 mg/kg) or pembrolizumab (25 mg/kg) as single agents or murlentamab/pembrolizumab combo-therapy. Data are represented as boxplots. *** *p* < 0.001, **** *p* < 0.0001 in comparison to baseline. (**E**) In vivo tumor growth. Data are represented as mean + SEM.

**Table 1 cancers-13-01845-t001:** Flow cytometry antibodies used.

Antibodies	Supplier	Catalog No.	Clone No.	Experiment
CD4-AF700	Thermofisher	56-0049-42	RPA-T4	CD69^+^ detection
CD15-eF450	Thermofisher	48-0158-42	MMA	
CD16-PC7	Thermofisher	25-0168-42	CB16	
CD25-PerCP	Thermofisher	46-0257-42	CD45-4E3	
CD56-FITC	Thermofisher	61-0567-42	CMSSB	
CD64-PerCPeF710	Thermofisher	46-0649-42	10.1	
CD69-PE	Thermofisher	12-0699-42	FN50	
CD127-APC	Thermofisher	17-1278-42	eBioRD5	
CD3-VB	Miltenyi	130-133-133	BW264/56	
CD45-VG	Miltenyi	130-113-124	5B1	
CD14- AF714	Miltenyi	130-113-144	TUK4	
CD8-PEeF610	BD BioSciences	563919	SK1	
CD3-AA700	Beckman	B10823	UCHT1	ICOS^+^ detection
CD8-KrO	Beckman	B00067	B9.11	
CD4-PB	Beckman	A99020	RMO52	
HLA-DR-ECD	Beckman	A66330	3G8	
CD45RA-PE	AbD Serotec	MCA1075F	AT10	
ICOS-APC	Beckman	IM1239U	84H10	
PD1	Cell Signaling	13684S	EIL3N	In vivo experiment
CD45-BV421	BD BioSciences	563879	HI30	
CD3-FITC	Miltenyi	130-098-162	REA613	
CD14-PE	Miltenyi	130-100-676	REA599	
CD163-APC	Miltenyi	130-112-129	GHI:61.1	
CD86-Alexa700	BD BioSciences	561124	2331	
CD11b-APCVio770	Miltenyi	130-110-556	REA713	
Viobility^TM^ Fixable Dyes	Miltenyi	130-109-816	-	In vitro experiments
CD11b-FITC	Miltenyi	130-110-552	REA713	
CD163-PE	Miltenyi	130-112-286	REA812	
CD36-PEVio770	Miltenyi	130-110-742	REA770	
CD206-APC	Miltenyi	130-124-012	DCN 228	
CD64-PerCPVio770	Miltenyi	130-116-303	REA978	
CD80-PE	Miltenyi	130-123-253	REA661	
CD32-PEVio770	Miltenyi	130-097-506	2E1	
CD282-APC	Miltenyi	130-120-138	REA109	
CD14-APCVio770	Miltenyi	130-110-552	REA599	
CD183-APC	Miltenyi	130-120-450	REA232	
CD3-APCVio770	Miltenyi	130-113-136	REA613	
CD4-VioBright FITC	Miltenyi	130-113-791	REA623	
CD25-PE	Miltenyi	130-115-534	REA945	
CD8-PEVio770	Miltenyi	130-110-680	REA734	
CD45-VioGreen	Miltenyi	130-110-638	REA747	

## Data Availability

All data relevant to the study are included in the article or uploaded as Appendix A.

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
