# Peer review of "Murlentamab, a Low Fucosylated Anti-Müllerian Hormone Type II Receptor (AMHRII) Antibody, Exhibits Anti-Tumor Activity through Tumor-Associated Macrophage Reprogrammation and T Cell Activation"

_cancers, 2021, doi:10.3390/cancers13081845_

Round 1

Reviewer 1 Report

There is indeed both novelty and merit to these findings regarding anti-AMHRII promoting anti-tumor TME via M1 macrophage polarization and ADCC. These findings are first noted with the use of ex vivo CRC and ovarian cancer patient samples from phase I/II trials. From there these findings are supported via in vitro assays demonstrating ADCC in using ovarian tumor cell lines and healthy donor macrophages induced for a TAM phenotype in both naive and TAM like to an M1 polarization. There is also an additive effect with anti-PD1 therapy in these assays.

These findings mainly limited by the need for additional mechanistic in vivo confirmation of these findings as figure 5D is the main limitation of the study along with the lack of exposition and evaluation in the additive effect of anti-PD1 and anti-AMHRII from the in vivo model and samples further. Additional confirmation and further evaluation of anti-PD1 and anti-AMHRII in the in vivo model would support this study significantly. Specifically, evaluation of T cell exhaustion and increased reversal with the addition of anti-AMHRII with anti-PD1.

Reviewer 2 Report

In this study, the authors have shown that murlentamab antibody reprograms both naïve and tumor-associated macrophages (TAMs) into M1 phenotype, polarizes CD4 T cells towards Th1 phenotype, and activates CD8 T cells, leading to anti-tumor activity. Murlentamab has already been in clinical trial and authors show that this treatment results in activation of both innate and adaptive immune response in patients with advanced or metastatic gynecological cancers and colorectal cancer. Authors also suggest that anti-tumor effects of murlentamab can be enhanced by combining with anti-PD-1. The findings from this study thus have important implications in cancer immunotherapy. However, some major concerns asoutlined below need to be addressed:

  1. Authors have shown that the co-culture of TAM-like MDMs with SKOV3-R2+ cells opsonized with murlentamab and 3C23K-CHO enhance the proportion of Th1 CD4+ T cells while lowering the proportion of Th2 CD4+ T cells. This also results in a strong reduction of regulatory T cells (Tregs). However, CD25 is not a sufficient marker of Tregs, therefore it is essential to show Foxp3 and may be TGF-beta and IL-6 production. Also, M1 macrophagesproduce IL-6 and IL-23, which expand the Th1 and Th17 cell In addition, reduction in Tregs is known to be balanced by an increase in Th17. Therefore, it is important to determine the status of Th17 subtype after murlentamab treatment.
  2. Authors combine murlentamab with pembrolizumab with an intent to enhance its anti-tumor effects. However, determination of expression of PD-1/PD-L1 on T cells and tumor cells is not shown. This would have provided a rationale to combine these and also would have helped to explain their findings. In this experiment, authors should also compare murlentamab with 3C23K-CHO antibody to show if less fucosylation is more effective than normal fucosylation of the antibodies. Authors should also show the profile of other immune cells in vivo (not only CD86+ and CD163+ cells).
  3. Although the polarization of macrophages towards M1 phenotype has been shown after murlentamab treatment, it is important to show the Immunosuppressive effects of macrophages before and after antibody treatment (also with anti-PD-1). This would also help in explaining the observed activation of T cells.
  4. There are inconsistencies in Figure S4 and Figure 3B: in S4, IL6 is not increased with M0 but in 3B, there is an increase; in S4, CCL4 is increased with TAMs but not in 3B.
  5. There are discrepancies in the data presented in the figures, figure legends, and in the text. For example, Figure 1A shows data at D3 and D4 and in the legend it is mentioned that cell numbers were determined after 1 and 2 days of co-culture, however, co-culture was started at day 3 as shown in Figure S2. Therefore, data should be at D4 and D5 (not at D3 and D4). Figure 3B: authors write that “co-culture of murlentamab-opsonized SKOV3-R2+ tumor cells with both M0 and TAM-like MDMs induced a significant increase in IL1β, TNFα, IL6 and IFN-gamma pro-inflammatory cytokine production”, however, TNF-alpha does not increase either with M0 or with TAMs. In the legend for Figure S4 authors mention that cytokines (IL-1β, IL-12, TNF-α, IL-6, IFN-gamma, IL-10) are measured, however, in the figure only IL12 and IL6 are shown. In the same figure legend, it is mentioned that chemokines (CCL-2, CCL-4, CCL-5, CXCL-9 and CXCL-10) are determined but CCL2 is not shown in the figure. Figure S5: 3C23K-CHO antibody was not used but is mentioned in its legend; also it is important to describe in the figure legend how Th1, Th2 and CD8 T cell activation were determined. Figure 5A-C legend mentions 3C23K-CHO antibody, however, in the figures this group is not shown.

Also, statistics is missing in some figures, for example for Figure 1B-D. Figure 2A: the authors mention that “the number of SKOV3-R2+ tumor cells opsonized with the 3C23K-FcKO control increased over time”, however, it does not appear to be statistically significant. It appears that with both 3C23K-CHO and 3C23K-FcKO there is no change over time (statistical significance is not shown for D0 vs D3 vs D4). Also, it is not clear in Figure 1C, how many patients out of total number of patients demonstrate changes in CXCL9 and CXCL10. For Figure 1A, it is not clear between which groups statistical comparisons are shown; important to explain what * and # represent in the legends. For Figure 5B-C: statistical comparisons should be made and shown for same antibody with and without pembrolizumab, for example for murlentamab without anti-PD-1 to murlentamab with anti-PD-1. Statistics should be provided for Figure 5E - tumor growth analysis.

In addition, minor concerns and suggestions are:

  1. Page 6, line 223: “diminution” should be replaced with “increase” of cells expressing M1…
  2. Supplementary Figure S3: In TNF-alpha and IL6, last bar and middle bar respectively are not visible; symbol for alpha is not proper in TNF-alpha.
  3. Figure 5C is not referred either in the text of the results (line 329, page 9) or in the figure legend.
  4. Line 330: Interestingly,…….. (Figure 5B). – this sentence is redundant as the previous sentence on line 329 conveys the same information.
  5. In Figure 5E, dose of murlentamab is shown as 5mg/ml- should be corrected.
  6. Figure 5B: Is the second bar graph Th2 CD4+ T cells (not Th1)?
  7. Figure 5D: it is not clear from which treatment group CD86+ and CD163+ cells are shown.
  8. Page 12, line 444: authors should include NK cell data.

Reviewer 3 Report

General comments:

The article attempts to provide the description of the mode of action of Murlentamab, a humanized glyco-engineered anti-AMHRII monoclonal antibody currently in clinical trials.

This is definitely highly interesting in the context of the initial phase I and II trial results with this novel anti-cancer treatment: “pilot study suggests longer than expected PFS for murlentamab and FTD/TPI in advanced mCRC, especially in patients with high AMHRII expression.”

However, the data description is very limited and unclear in places. Most importantly, the data validity is questionable (see the comments below in bold).

The figures are blurry and the quality should be improved.

Statistical methods single data points and sample numbers should be included in each figure and the respective figure legends, p value for each comparison instead of * or # etc.

Specific comments:

Fig 1. should also show the data for non-Treg cells, for a more comprehensive overview of the treatment effects

Representative flow cytometric plots and histology should be provided either in the main figure or in the supplement as for part E. Gating strategy for flow cytometry should be included in the supplement. Part E is lacking the description of the colours. Some figures are very low quality (grainy), which should be improved.

Figure 3 results section 2.3 provides a very nice overview of what methods were used for each experiment “M1 or M2 markers was evaluated by flow cytometry and their ability to produce pro- and anti-inflammatory mediators was assessed by AlphaLisa immunoassays”. This should also be included in the Results section 2.1.

Fig 2. Again the figure quality is underwhelming and the p values should be shown instead of *-****

Statements such as “Antibody-Dependent Cell-mediated Cytotoxicity (ADCC) is a well described function of macrophages and NK cells that has been attributed to the anti-tumor activity of several antibodies used for cancer immunotherapy.”, should provide references.  

Fig 3. As for Fig 1, representative flow cytometric plots should be shown to strengthen the validity of data. This is especially relevant in the context of CD32 expression:

  1. The authors do not describe what solid parts of the bars mean. This should be included for all the figure legends.
  2. If my interpretation of the bar ranges is correct the top part of the bar should indicate the max value observed within the experiment. Provided this interpretation is correct, this raises some very serious concerns regarding data analyses/interpretation as for the bar graph showing CD32 expression the y axis represents percentage of positive cells, which can not go above 100%, while it can be clearly seen for the 3C24K-CHO condition and potentially for Murlentamab condition that the top part of the bar goes above 100%. In the light of this, I would like the authors to (1) explain how this was achieved? (2) provide all the data for every data point of the relevant figure (3) change the graph types for ALL the figures to show single data points.

I think at this point it is clear to me that the authors should perform some serious data revisions, representation changes and additional data description in all the figure legends to prove the validity of their data.

Some additional comments:

In the results section the authors state: “In line with previous results, the culture of TAMs with the murlentamab antibody induced a significant increase in TNFα and IL-6 pro-inflammatory cytokines”. To start with, this is not completely true as there was no significant increase in TNF production in Fig 3B. Moreover, the y axes of the Fig S3 should be harmonized with the main figure and show pg/ml to simplify the comparison of the data.

Fig S3 has excellent quality and the authors should strive to provide the same quality in the main figures.

A table describing all the Flow cytometric reagents, their clones, product numbers and vendors should be provided in the M&M section.

English language quality is not great (e.g. “For detecting CD69+ cells in C201 study, 100μl of fresh total blood was used for each staining and incubate first for 20 minutes on ice with 20μl of anti-human Fc receptor binding inhibitor.)

Round 2

Reviewer 2 Report

Authors have answered most of my concerns. A few concerns need to be addressed:

  1. Figure S6 and S7: Please change “Lived cells” to “Live cells”.
  2. In the graphical abstract- change “biopsie” to either “biopsies” or “biopsy”.
  3. Figure 5D legend: “Quantification of circulating CD86+ and CD163+ cells by flow cytometry from blood of tumor-bearing mice before treatment and after 24 days of treatment with murlentamab as single agent….” Authors should add information about the new data, that is pembrolizumab treatment as well as murlentamab plus pembrolizumab treatment in the legend.
  4. In response to my earlier comment: “Figure S5: 3C23K-CHO antibody was not used but is mentioned in its legend; also it is important to describe in the figure legend how Th1, Th2 and CD8 T cell activation were determined”, authors have responded that “Moreover, as suggested by the reviewer and in order to homogenize all the figures, we added in the histograms (y-axis) of the supplementary Figure S5 and Figure 4 the markers used to determine Th1/Th2 polarization and CD8+ T cell activation.”

However, for Figure S5A, the markers for Th1 and Th2 are not correct (on y-axis); CD183+ cells should be only for Th1 population (not for Th2).

  1. In light of the p-values provided by the authors for figure 2A, this statement on lines 202-203: “the number of murlentamab-opsonized SKOV3-R2+ significantly decreased over time both in presence of M0 or TAM-like MDMs (Figure 2A)”, is not true. p-values for D3 vs D4, D3 vs D5, and D4 vs D5 are not significant for both M0 or TAM-like MDMs. Authors need to be careful, there are differences with respect to 3C23K-FcKO (as shown by stars) but not over time; therefore, this statement needs to be corrected.
  2. Not sure if any formatting problem but this concern still remains: Supplementary Figure S3: In TNF-alpha and IL10, last bar and middle bar respectively are not visible; symbol for alpha is not proper in TNF-alpha.
  3. For my earlier comment: Figure 5C is not referred either in the text of the results (line 329, page 9) or in the figure legend, authors have responded: “This Figure 5C is now referred page 12, line 397 and line 401”. However, 5C is still not referred in the figure legend.

Reviewer 3 Report

General comments:

The authors definitely did a great job revising the manuscript and addressing the issues. This is especially evident in the improvement of Figures 2-5, as well as providing all the p values, a table of flow cytometry antibodies in M&M and particularly reconciling the data from the figure (Figure 3A) I had most concerns with.

Even though I think currently the trend is towards putting the actual p values on the graphs, I thank the authors for providing those and will leave it with the journal to pursue if necessary.

I am happy and thank the authors for providing detailed gating strategies and fully understand the challenges of multicentric studies like this one.

Specific comments:

The graphical abstract and images in Fig 1E and F are still low quality and blurry upon zooming in. Moreover, in Figure 1F (graph 1) authors once again seem to have problems with their percentages and have the scale going from 160 to 280 % of cells. I assume they just got lucky with the cell numbers in the second plot, which is in a possible percentage range. I do believe that it is a genuine mistake and the y axis for both graphs should be something along the lines of “number of cells”. Please also check the y axis names in Fig 1E. Furthermore, the naming seems to run onto itself in Fig 1F graph 2.

I still have a problem (apart from the quality) with the images in Fig. 1F. Firstly, there’s definitely a red spot in right graph (towards top left), which is not annotated, which I assume is CD8 as in Fig. 1E. However, despite not being colour blind I honestly do not see any overlap of pink/magenta (CD16+) and cyan (Granzyme B) staining even in the right plot. Therefore, I would ask the authors to either provide a convincing higher magnification image in the corner of the current picture(s) or provide single staining pictures before the final overlay picture, which is currently on display.
